# An Overview of Transboundary Animal Diseases of Viral Origin in South Asia: What Needs to Be Done?

**DOI:** 10.3390/vetsci9110586

**Published:** 2022-10-24

**Authors:** Gyanendra Gongal, Habibar Rahman, Kishan Chand Thakuri, Kennady Vijayalakshmy

**Affiliations:** 1Veterinary Epidemiologist, Veterinary Epidemiology Centre, Directorate of Animal Health, Kathmandu 44600, Nepal; 2International Livestock Research Institute, South Asia Office, New Delhi 110012, India; 3National Laboratory for Foot-and-Mouth Disease and TADs, Directorate of Animal Health, Kathmandu 44600, Nepal

**Keywords:** cross-border, eradication, livestock, south Asia, trade, transboundary animal diseases, virus

## Abstract

**Simple Summary:**

The Transboundary Animal Diseases (TADs) are highly transmissible epidemic diseases of livestock which have the capability for rapid spread to new areas and regions regardless of national borders. The TADs are a major threat to livestock of any nation as they have the potential to cause large-scale damage, staking the food security of the country, and can cripple the nation’s economy significantly by direct loss in the form of disease conditions and deaths in affected population or indirectly due to required counter epizootic measures, loss in trade and probable zoonotic transmission. South Asian countries are more vulnerable to the introduction of Transboundary Animal Diseases (TADs) because people continue to move commodities and animals across borders. In addition to South Asian countries, TADs like Peste des Petits Ruminants, Avian Influenza, Foot-and-Mouth Disease, Lumpy Skin Disease and African Swine Fever are generating significant economic losses worldwide. Livestock diseases can result in significant losses for farmers’ subsistence and have a negative impact on the nation’s economy. The existing policies need a review in the light of recent understanding of drivers of emergence and re-emergence of transboundary animal diseases. Hence, there is a need for a critical review of the existing policies that regulate the preparedness for the prevention of TADs. Rapid diagnosis, surveillance, immunisation, and coordination among all the South Asian countries. The present study focuses on the state of several TADs in South Asia and provides plans of action and suggestions that the scientific community and authorities on animal health might implement.

**Abstract:**

Transboundary animal diseases (TADs) pose a serious threat to the sustainability and economic viability of the existing animal agriculture ecosystem in south Asia. The rapid spread of African swine fever and lumpy skin diseases in south Asia must be considered a wake-up call to prevent the entry, spread, and establishment of new exotic TADs, as south Asia has the highest density of livestock populations, and it will have a huge socioeconomic impact. Regional cooperation for the prevention and control of TADs is necessary, but rational decisions should be made to initiate even sub-regional cooperation in the present geopolitical situation. Cross-border collaboration for surveillance, early warnings, and animal movement control should be encouraged on a bilateral or multilateral basis as many countries share a porous border. Foot-and-mouth disease (FMD), peste des petits ruminants (PPR), and avian influenza (AI) have been identified as regional priority TADs, and many regional and country initiatives have been undertaken in the last two decades that need to be translated into action. The incursion of exotic TADs into south Asia has compelled us to rethink overall policies and strategies for prevention and control of TADs. This paper took into consideration six emerging and endemic TADs of viral origin to suggest a future course of action.

## 1. Introduction

TADs are defined by The Food and Agriculture Organisation of the United Nations (FAO) as highly contagious and transmissible epidemic diseases of livestock which have the capability for rapid spread to new areas and regions, regardless of national borders, and have serious socio-economic and public health consequences [1]. Transboundary animal diseases are highly contagious and have the potential for rapid spread, irrespective of national borders, causing serious socioeconomic consequences [2].

The south Asian countries have established a regional specialised organisation called the South Asian Association for Regional Cooperation (SAARC) in 1985, which comprises Afghanistan, Bangladesh, Bhutan, India, the Maldives, Nepal, Pakistan, and Sri Lanka. Contribution of livestock to agricultural gross domestic product (GDP) varies between 13 and 61%. SAARC countries represent around 20% of the global small ruminant population [3]. Average densities of livestock in this region are the highest in the world, and India possesses the largest amount of livestock in the world. As compared to Europe and America, livestock cannot be seen only from a productivity point of view, but they have a socio-cultural value, and they are also known as living banks. Since the Global Framework for Progressive Control of TADs (GF-TADs) has identified the SAARC region for sub-regional activity in Asia, eight SAARC countries were included as part of the South Asia region. Only six south Asian countries share a long porous land border with neighbouring countries, including the People’s Republic of China. Although Sri Lanka and the Maldives are island countries in south Asia, there is a geographical barrier, but they are not risk-free for TADs.

High-impact animal diseases of viral origin such as FMD, PPR, Classical Swine Fever (CSF), or African Swine Fever (ASF), while not directly affecting human health, do affect food and nutrition security, livelihood, livestock production, and trade [1]. The FAO and World Organisation for Animal Health (WOAH) have been working together for progressive control of TADs, particularly in low- and middle-income countries. So far, rinderpest is one of the TADs which has been eradicated in 2010 through the global rinderpest eradication programme under FAO/WAHO leadership. There is proof of the concept that TADs can only be eradicated through regionally coordinated campaigns and international partnerships.

The TADs are a major threat to livestock of any nation as they have the potential to cause large-scale damage, staking the food security of the country, and can cripple the nation’s economy significantly by direct loss in the form of disease morbidity and mortality in an affected population or indirectly due to required counter epizootic measures, loss in trade, and probable zoonotic transmission. The emergence of TAD poses a challenge to the country’s preparedness and mitigation policies. The existing policies need a review in light of recent understanding of drivers of emergence and re-emergence of transboundary animal diseases. Hence, there is a need for a critical review of the existing policies that regulate the preparedness for the prevention of transboundary infectious diseases of livestock and poultry. The present review covers the current situation of different TADs in the south Asia region and details some plans of action and recommendations that will be useful for the scientific community and animal health authorities.

## 2. Materials and Methods

The literature review was carried out for eight SAARC member countries, namely, Afghanistan, Bangladesh, Bhutan, India, the Maldives, Nepal, Pakistan, and Sri Lanka. Technical reports from the World Health Organization (WHO), World Organization for Animal Health (WOAH, formerly known as OIE), FAO, SAARC Secretariat, and World Bank websites were included in the review. The government websites and published reports from the respective countries were also considered and reviewed. Search engines such as Google Scholar and Science Direct were also used. Relevant scientific literature was searched using international electronic databases such as “PubMed”. For the present study, more than 100 published papers and documents were considered. We conducted a systematic review to cover all the aspects of TAD in the region. We performed the review in accordance with the PRISMA Statement methodology checklist depicted in Figure 1.

## 3. Results

### 3.1. Situation Update of Priority TADs and Progressive Control of TADs in South Asia

The present review was restricted to six transboundary animal diseases (TADs) of viral origin, namely, FMD, PPR, lumpy skin disease (LSD), ASF, CSF, and highly pathogenic avian influenza (HPAI). These TADs are notifiable diseases under the WOAH (OIE), and most countries in south Asia have identified them as notifiable animal diseases. The status of major TADs of viral origin in SAARC member countries is presented in Figure 2.

#### 3.1.1. Peste des Petits Ruminants 

PPR is an acute, highly contagious, and fatal disease primarily affecting goats and sheep. It was reported for the first time in 1989 in south Asia and it is endemic in all SAARC countries except Sri Lanka, where the disease has never been reported [4,5,6,7]. Each year, PPR causes economic losses worth an estimated USD 1.2 to 1.7 billion, due to animal deaths, reduced production, and the cost of fighting the disease [3]. Approximately one-quarter of the budgetary impact occurs in south Asia [8]. Considering the impact of PPR in the livelihoods of populations, particularly in rural areas, India, Bangladesh, and Nepal started PPR vaccine production as a part of a PPR control programme. The current PPR vaccine has two major constraints: thermolability and inability to differentiate the infected from vaccinated animals [9].

ILRI has identified PPR as one of the priority animal diseases whose control should be considered for poverty alleviation in Western Africa and south Asia, which highlights the economic importance of PPR [10]. The WOAH and the FAO, in their joint global strategy for control and eradication of PPR, have set the goal of eradicating this disease by 2030. The GF-TADs with the involvement of SAARC countries developed a regional roadmap for PPR eradication in south Asia, and each country has been encouraged to develop a national strategic plan for PPR. The second PPR Regional Roadmap Meeting for SAARC countries was held in Dhaka in 2018 and reviewed PPR stage progression 2018–2030 on the basis of country self-assessment, as shown in Table 1, and possible planning and implementation of the PPR regional roadmap [3].

The meeting recommended carrying out an assessment using the PPR Monitoring and Assessment Tool (PMAT), which will contribute to identifying PPR risk areas and practices along with the small ruminant value chains that may contribute to PPR introduction and/or spread to formulate targeted eradication plans that efficiently eliminate the disease.

Since PPR vaccination remains the main tool to control PPR outbreaks in endemic countries, it was also recommended that PPR vaccination should be time limited (two successive years of vaccination in Stage 2, followed by vaccination of young animals (4 months to one year in age) within one year or two), with high coverage aiming for 100 per cent vaccination coverage to achieve the necessary flock immunity in high-risk areas.

#### 3.1.2. Foot-and-Mouth Disease

FMD is the most important TAD in south Asia as it affects mainly the cattle, buffalo, and pig population but also infects sheep, goats, and captive and free-range wildlife populations [11,12]. The disease is highly contagious, and the potential for infection of different domesticated and wildlife hosts, not all of which show obvious signs of disease, is a further challenge to control [13]. Goats are the maintenance hosts; pigs are the amplifying hosts, and cattle are the indicator hosts for the FMD virus (FMDV) [8]. The FMDV is transmitted most efficiently via airborne or aerosol spread, especially when animals are in close contact [14]. The high occurrence of FMD outbreaks is seasonal, and it could be due to excessive movement of food animals for agricultural activities as well as unrestricted importation of large numbers of animals across the borders during religious festivals [15].

FMDV serotypes O, A, C, and Asia-1 were circulating in south Asia in the past, and quadrivalent FMD vaccines were used until the year 2000. At present, only three serotypes (O, A, and Asia1) of FMDV are circulating in the livestock population in south Asia, and serotype O was responsible for 80% of the confirmed outbreaks [15,16,17,18,19]. Unrestricted animal movements in cross-border areas are determining factors for circulation of different serotypes of FMDV in south Asia.

New virus strains evolve and emerge, invariably leading to large outbreaks and challenging the efficacy of FMD vaccination. Vaccination with killed vaccines is used on a large scale, but the immunity induced is short lived and is serotype and sometimes strain specific, and vaccination does not always confer protection against all strains within the serotype [20,21]. On the other hand, if vaccinated animals are partially protected, they may be able to support viral replication, thus posing a risk of infection to other animals [22]. As diagnosis and slaughter policy cannot be practised in India and Nepal (due to ethical and socio-economic reasons) and animal movement control is not feasible, routine vaccination is the best way to achieve protective antibody response against FMD in the vaccinated animals [23,24]. The quality and efficacy of the FMD vaccine is an issue that may be linked to low antigen levels, poor inactivation, and mismatching of vaccine strain with a field strain of the FMD virus. FMD vaccines need to be of superior quality, which can confer higher levels of protection by eliciting neutralising antibodies; quality and potency of FMD vaccines commercially available in south Asia is a major concern.

A progressive control pathway for FMD (PCP-FMD) was established to determine national progress and to develop national and regional action plans and support, which has been adopted as a joint tool between FAO/EuFMD/OIE [25,26].

#### 3.1.3. Lumpy Skin Disease

LSD is primarily a vector-borne disease of cattle and buffaloes, but it can be transmitted through direct or indirect contact and artificial insemination. Historically, LSD was confined to Africa and part of West Asia until the 1980s, but it has been rapidly spreading to newer territories since 2015. The main pathways for LSD introduction into free areas were considered to be the movement of infected animals and vectors. The economic impact of LSD for southern, eastern, and south-eastern countries was estimated to be up to USD 1.45 billion in direct losses of livestock and production [27].

LSD is an emerging TAD in south Asia. LSD has spread to Afghanistan, Bangladesh, Bhutan, India, Nepal, and Pakistan in the last 3 years, and it clearly demonstrates it is linked to uncontrolled cross-border movements of animals through porous borders [28,29,30,31,32]. The LSD outbreaks have been reported from remote and isolated island territories such as the Andaman and Nicobar in India, which is a matter of concern for island countries. FAO conducted a qualitative risk assessment of the likelihood of introduction and/or spread of LSD in 23 countries in South, East, and Southeast Asia covering the period October–December 2020, and the risk map of LSD is presented in Figure 3 [27].

It was observed that vaccination was most effective (40 percent) if protection had already been developed at the time of virus entry, more so than large-scale vaccination after virus entry [33]. Considering the rapid spread of LSD in India affecting cattle and buffalo populations, the ICAR-National Research Centre on Equines (ICAR-NRCE), Hisar (Haryana), in collaboration with the ICAR-Indian Veterinary Research Institute (IVRI), Izatnagar, Uttar Pradesh, has developed a homologous live-attenuated LSD vaccine “Lumpi-ProVacInd” [34]. Sheep pox virus (SPV) and goat pox virus (GPV)-based vaccine (heterologous vaccine) is usually authorised to induce cross-protection against LSD in cattle, wherein the homologous LSD vaccine is not available but heterologous vaccines may not provide complete immunity [35].

#### 3.1.4. Classical Swine Fever

CSF is a transboundary disease of wild and domestic swine thought to be enzootic in south Asia. The CSF remains a threat to pigs raised in south Asia, including Bangladesh, Bhutan, India, and Nepal [36,37,38,39].

It should be noted that the CSF virus can cross the placenta and infect the developing foetus, leading to persistent infection, particularly during mid-gestation, which is not the case in ASF [40,41].

The CSF is a notifiable disease in Bhutan [42], India, and Nepal, and these countries have CSF prevention and control measures in place, including available locally produced CSF vaccines [43]. Pig farmers in the North-East region of India often import the CSF vaccine from Myanmar as there is a high density of their pig population and pig meat is in high demand. The lapinised CSF vaccine is generally used in post-outbreak situations as it provides early protection against virulent strains, even at 1 day post-vaccination and the administration of a single dose [44]. Since live attenuated vaccines elicit a multivalent immune response and vaccinated pigs cannot be distinguished from infected pigs by serological methods (DIVA), there are trade restrictions on animals from areas practising vaccination with these strains [45].

#### 3.1.5. African Swine Fever

ASF, a highly contagious viral disease that affects both domestic and wild pigs, is usually fatal, with mortality reaching almost 100% [46]. The remarked feature of the ASF virus is the long-term persistence and viability of the virus in contaminated fomites, foods, and feeds originating from infected pigs, which is an extremely favourable basis as a human-driven disease [47]. Small holder piggery farmers are the most vulnerable to the economic losses due to African swine fever [48]. Although many ASF cases had been reported in wild boar in Europe and East Asia, ASF outbreaks in most Asian countries have been mainly associated with domestic pigs [49]. The latest update on the ASF situation is depicted in Figure 4.

India reported its first outbreak of ASF in the Northeastern states of the country in 2020. Bhutan and Nepal reported an ASF outbreak in 2022.

The early detection and eradication in ASF-free zones is vital but it may be difficult in areas where backyard farming of pigs is predominant. The control of the disease is hampered by the lack of an effective vaccine and treatment, being limited to strict sanitary measures including the slaughter of affected animals and closure of trade borders, which implies important economic losses [51].

#### 3.1.6. Highly Pathogenic Avian Influenza

HPAI is one of the economically important poultry diseases of zoonotic potential and it has drawn international attention since 2003 due to the public health threat of pandemic. The HPAI outbreaks have a serious impact on livelihoods, the economy, and international trade in affected countries. The HPAI (H5N1) has affected almost all countries, except a few island countries in Asia. The FAO, OIE, and WHO provided a greater role in capacity building and resource mobilisation.

HPAI viruses fall into groups with H5 and H7 hemagglutinin subtypes and may result in 100 percent mortality, but not all H5 and H7 viruses cause HPAI [52]. Outbreaks of HPAI (H5N1) in the poultry population have been reported in all SAARC countries except Sri Lanka and the Maldives. HPAI outbreak in the poultry population is continuing in south Asia, as shown in Figure 5.

According to the WHO, a total of 13 human cases of AI (H5N1) have been reported from Bangladesh, India, Nepal, and Pakistan during 2007–2021. The cumulative number of confirmed human cases for avian influenza A(H5N1) reported to WHO during 2003–2021 is presented in Table 2.

H9N2 avian influenza viruses have become globally widespread in poultry over the last two decades and represent a genuine threat both to the global poultry industry but also humans through their high rates of zoonotic infection and pandemic potential [55]. Since poultry vaccination has been implemented using the AI (H9N2) subtype for decades, human cases of AI (H9N2) have been reported from all major poultry-production-based countries. H9N2 viruses are considered endemic in Bangladesh and Pakistan, and are likely endemic in regions of Afghanistan, India, and Nepal [56,57,58,59]. According to the WHO, Bangladesh, India, and Pakistan have reported human cases of AI (H9N2) during the last one decade. The phylogeographic range of poultry-adapted H9N2 lineages is presented in Figure 6.

The phylogeographic range of poultry-adapted H9N2 lineages, as shown in Figure 6, clearly indicates that G1 ‘Western’ viruses constitute the majority of viruses found in poultry in South Asia, the Middle East, and Northern Africa.

### 3.2. Regional Initiatives and Challenges

The establishment of the SAARC by Bangladesh, Bhutan, India, the Maldives, Nepal, Pakistan, and Sri Lanka in 1985 was a strategic and bold decision to promote regionally coordinated activities to move forward a socioeconomic development agenda, including harnessing natural resources [60]. Afghanistan joined the SAARC in 2005. The SAARC Agriculture Centre based in Dhaka is a specialised body to cover livestock and poultry diseases of regional importance. Although institutional mechanisms for regional cooperation have been well established and driven by SAARC member states, implementation of policy decisions is hampered by geopolitical crises and lack of political will from time to time.

Rinderpest was one of the TADs that was targeted for eradication campaigns through regionally coordinated initiatives in South Asia. There was an attempt to launch the South Asia Rinderpest Eradication Campaign (SAREC) during the 1980s, which was unable to be materialised due to geopolitical reasons [61,62]. The European Union supported implementation of OIE pathway for rinderpest eradication, including submission of a dossier for rinderpest-free status through the Strengthening Veterinary Services and Livestock Disease Control (SVSLDC) project in five countries of the SAARC region [62].

### 3.3. Global Framework for the Progressive Control of Transboundary Animal Diseases (GF-TADs)

The GF-TADs are a joint initiative of the FAO and WOAH (OIE) to achieve the prevention, detection, and control of transboundary animal diseases (TADs), designed to empower regional alliances in the fight against TADs in order to provide capacity building and to assist in establishing programmes for the specific control of certain TADs on the basis of regional priorities [63]. There have been resurgence and outbreaks of many high-impact TADs in the Asia–Pacific region, including FMD, HPAI, CSF, porcine reproductive and respiratory syndrome (PRRS), PPR, and rabies. The Regional Action Plan addresses the animal diseases and topics that have qualified as a ‘priority’ for the region. The SAARC region had identified HPAI, FMD, and PPR as priority TADs; the Regional Support Unit has been established in Kathmandu, while leading laboratories for diagnosis of HPAI, FMD, and PPR are located in Islamabad (Pakistan), Mukteshwar (India), and Dhaka (Bangladesh), respectively [63].

Key achievements of regional GF-TADs for South Asia (2005–2011) are summarised in Table 3.

The Emergency Centre for Transboundary Animal Diseases (ECTAD) was established at the global level by FAO in the wake of the avian influenza crisis in 2004, which has been regionalised as enabling countries in the region to better prevent and control the emergence and spread of TADs, zoonotic influenza, and other zoonotic emerging infectious diseases (EIDs) at national and regional levels. The ECTAD has been established under FAO country offices in Bangladesh and Nepal, playing an important role in strengthening surveillance, laboratory diagnostic capacity, and prevention and control of TADs, including One Health coordination for zoonoses.

### 3.4. Performance of Veterinary Services (PVS) in SAARC Countries

The World Organisation for Animal Health (WOAH), previously known as the OIE, has an institutionalised evaluation of the performance of national veterinary services in its member states, considering the key role of the veterinary services in ensuring the quality of veterinary services, including surveillance, prevention, and control of TADs and zoonoses. The OIE PVS evaluation using the PVS tool is a mechanism to ensure full coverage of the veterinary domain and is primarily used to identify areas of relative strength and weakness within a particular national veterinary service, against relevant international standards [64]. The OIE has developed its own forms of PVS targeted support to assist countries in key areas such as One Health collaboration, veterinary legislation, laboratories, education, and OIE focal points. All SAARC countries have conducted a PVS evaluation at least, and India was the last one, as shown in Table 4.

The evaluation of the performance of veterinary services is important for TADs as recognition of declaration of disease-free status for TADs, addressing a number of issues of critical importance to veterinary services such as the implementation of standards, laboratory biosafety and biosecurity, disease investigation and tracing, livestock markets (domestic trade), and public private partnerships. Unfortunately, PVS and PVS gap analyses are not updated, as shown in Table 4.

### 3.5. Determinants and Drivers for TADs

#### 3.5.1. Population, Density, and Farming Practices

South Asia covers about 5.2 million sq. km, which is about 3.5% of the world’s land surface area. The human population of South Asia is about 1.92 billion, which is about one-fourth of the world’s population, making it the most densely populated geographical region of the world. High population size and density drives high demand for animal protein and cereals, as well as high frequency of animal trade and movement. South Asia is also complemented by greater biodiversity, agro-ecological conditions, and large livestock and poultry populations. There are about 438.65 million cattle/buffaloes, 396.77 million sheep/goats, and 2765.50 million poultry [65]. As a result, average densities of livestock in this region are the highest in the world, i.e., 70–137 per square km. The livestock sector is the fastest growing and emerging sector in agriculture in South Asia due to suitable agro-ecological conditions, high-yielding indigenous breeds of livestock leading to comparatively low cost of production, ever-growing domestic demand for animal protein, incentives for export of livestock products, etc.

Artificial insemination (AI) has been introduced for breed improvement, with European breeds of cattle in most countries; male calves born after AI are not fit for draft purpose nor for meat production, and they are deprived of cow milk and become part of the stray animal contingent. Male buffalo calves are considered a burden by farmers, and they are deprived of milk and thus rely on grass feeding.

The conventional practice of stamping out or test and slaughter policy cannot be executed in Bhutan, India, and Nepal for animal disease control as slaughter of cattle is prohibited. The stamping out policy is preferred for early eradication of infectious agents of interest and free access to international market in a short time period, but it is very costly to execute, and recent mass culling of animals in the name of containment and subsequent eradication of FMD, HPAI, and ASF has raised serious animal welfare concerns and resistance from farmers.

The encroachment of agricultural land for real estate development, deforestation, mining, scarcity of water and grazing land/pasture, replacement of manures with chemical fertiliser, and mechanisation of agricultural farming and rural transport have compelled farmers to abandon livestock and draft animals which have an impact on road safety and biosecurity measures to be taken for prevention of TADs.

#### 3.5.2. Traditional Livestock Market and Fair

Livestock fairs and markets have been institutionalised in several South Asian countries with land borders. These traditional livestock markets have existed for decades at village or district levels, being popularly known as weekly markets in rural and semi-urban settings for sale of goats, pigs, cattle, buffaloes, and poultry. Farmers and livestock traders have a good idea of weekly markets, and there is a congregation of livestock from various parts of the country and neighbouring countries. As a result, many TADs have been introduced or spread over time within and across borders.

#### 3.5.3. Transboundary Movement of Livestock and Poultry

Globalisation and growing demand of livestock and its products have accelerated movement of live animals and livestock products across the border, which has increased the potential risk of TADs and has facilitated TAD incursion into free areas. The most common ways of introduction of TADs to a new geographical territory are through importation of infected or diseased animals or contaminated animal products including animal production materials. There are instances of introduction of airborne and vector-borne TADs through air current or vectors such as ticks, mosquitoes, and gnats. Global warming and climate change may have a direct impact on propagation of vector-borne TADs of viral origin such as bluetongue and Rift valley fever. The African swine fever is another example of a viral TAD which has adapted in wild pigs and spread from Eastern Europe to Asia through movement of wild boars as well as transportation of contaminated pig products.

Cross-border value chain studies for poultry movement and trade were made by the FAO under USAID to better understand disease transmission dynamics in cross-border areas, but their value was limited for documentation and the recommendations were not translated into action. The development of activities such as distribution of goats to unemployed youths, a marginalised population, by the government and NGOs by importing goats from neighbouring countries have played a crucial role in the entry, spread, and establishment of PPR in new areas during the 1990s.

#### 3.5.4. Cross-Border Trade of Food Animals

There is a tendency to prevent legal and official export/import of livestock for preservation of genetic materials by some countries, but this only facilitated illegal movement of livestock animals in cross-border areas. Cross-border movement of cattle, goats, pigs, and poultry has facilitated entry and spread of FMD, PPR, LSD, ASF, CSF, and HPAI. There were several attempts to organise a series of consultations to establish cross-border collaboration for information sharing and surveillance of TADs at bilateral and multilateral platforms, but it did not lead to any concrete policy decision and action due to weak border control mechanisms; lack of understanding, trust, and confidence for cross-border collaboration; and limited dialogue at technical levels only. There is no veterinary sanitary inspection of meat, meat products, and animal production materials imported to South Asia, which may become a potential source of outbreaks of TADs due to persistence of viruses such as FMD, ASF, and LSD.

#### 3.5.5. Genetic Material and Live Animal Import Policy

There is a growing tendency to introduce exotic breeds of livestock animals to improve animal productivity to boost animal production in countries of South Asia. These exotic breeds of animals, particularly from Africa, without proper veterinary sanitary inspection and import risk analysis may pose a threat of introducing new exotic TADs in one country and then spread over other countries of South Asia. African swine fever and lumpy skin disease have been introduced in several countries in South Asia. If contagious caprine pleuropneumonia (CCPP) or any livestock diseases limited in Africa are to be reported in South Asian countries, no one should be surprised. The entry, spread, and establishment of any exotic TADs in one of the countries in South Asia may be a threat to the whole region due to informal movement and trade of live animals and products.

#### 3.5.6. Public Private Partnership

Livestock production and trade are managed by farmers, and the private sector and government sector facilitate them with surveillance, prevention, and control of economically important animal diseases and incentives to boost animal production. It is the responsibility of a competent veterinary authority to prevent incursion of TADs and to execute government policy, strategy, and action plans to control and subsequently eradicate TADs out of national interest. Private sectors should cooperate with competent veterinary authorities in ensuring veterinary sanitary inspection and quality control of products.

Animal traders often consider the profit margin in livestock trade as a major consideration, while they do not take disease risk factors due to ignorance and lack of awareness. It is important to build up trust and understanding between competent veterinary authority and animal traders so that they better understand their business and welfare is directly linked to prevention and control of TADs.

## 4. Discussion

The rinderpest eradication campaign launched during the 1970s in Nepal and neighbouring countries clearly demonstrated that mass vaccination and animal quarantine measures can bring down the number of cases towards zero, but that the disease may re-emerge after many years, unless there is a regionally coordinated programme in place [66].

Regional cooperation is a prerequisite for prevention, control, and subsequent eradication of viral TADs, and several initiatives have been taken by GF-TADs and other partners in South Asia.

Public private partnership is critical for early detection, prevention, and control of TADs in efficient and cost-effective ways, which is possible through confidence, trust, and understanding between public and private sectors. Although a competent veterinary authority is trying to prevent and control economically important TADs in the best interest of smallholder farmers and private sectors through regulatory and technical means, there is a challenge in terms of disease reporting, control measures, compensation, and perception. Biosecurity, diagnostics, and quality vaccines are critical needs for improving the poultry health management system in SAARC countries, which must be developed under public–private partnership.

There is great disparity across the region in the strength and resources of the animal health services and livestock sector, which may have an impact on TAD prevention and control. A country with a strong commercial sector, operating within a livestock sector that is important to GDP and export, is more likely to have political commitment to TAD control than one with a weak economy, poor tax base, and poorly organised producers [67]. Prevention, control, and subsequent eradication of any viral TADs may require multidimensional approaches which may include: (1) vaccination of the susceptible animal population with high vaccination coverage, i.e., >80 percent; (2) movement control and quarantine; and (3) slaughter campaigns (where feasible according to the country context).

Animal vaccination against TADs as disease control measures and access to international markets are trade dispute issues because of technological deficiency in vaccine production and surveillance. The availability of safe and effective DIVA (Differentiating Infected from Vaccinated Animals) vaccines and a corresponding laboratory diagnostic test could solve this dilemma if DIVA vaccination becomes an internationally accepted method for emergency vaccination without disruption of trade [45].

There are two dimensions of disease prevention and control programmes, i.e., bio-medical and socioeconomic. Poverty and ignorance are two of the factors driving people to eat sick or dead animals, as the majority of TADs are not zoonotic. The slaughter and sharing of meat and offal from sick and dead animals to neighbours and relatives in other villages or selling them at half price and the swill feeding practice have caused outbreaks of FMD. It continues to be a common factor for the introduction of ASF and CSF in countries previously free of it. In other words, TADs prevention and control measures should seriously consider human activity in outbreak areas as risk factors for disease transmission [68]. There is no provision of regulating or banning swill feeding, which may be responsible for accelerated transmission of ASF in Asian countries.

Considering the significance of regional cooperation and preparedness for effective prevention and control of TADs, a strategic consultation meeting was organised jointly by the National Academy of Agricultural Sciences (NAAS), ILRI, SAARC, and Bangladesh Academy of Agriculture (BAAG) on 15 February 2022 [69]. The following recommendations require due attention from policymakers, international partners, and private sectors.

On challenges and priorities of TAD in South Asian countries:Each country may prioritise three TADs initially, considering their disease severity, zoonotic threat, and trading to other countries, in order to develop a consensus among the member countries on addressing the priority diseases using both managemental and frontier science.Development of animal disease screening facilities at the major animal transportation route with appropriate diagnostic kits and trained manpower.Development of a SAARC vaccine bank for a quick response to any epidemic/pandemic.

On strategies to strengthen regional collaboration and funding for preparedness of TADs in the region:One TAD coordination centre for South Asian countries, preferably with the ILRI, may be established for better preparedness against TAD.A regional program on TAD, in the line of One Health, may be framed in consultation with WHO, WOAH (OIE), FAO, UNEP, and SAARC to complement and supplement the effort of each member country in controlling TAD.The Chief of the Animal Husbandry Department/Research Institution of each South Asian country needs to earmark a dedicated fund for managing and controlling TADs. Philanthropic organisations may also be approached for the cause.The efforts to contain TAD in the region may be continued through half-yearly meeting/workshop/brainstorming sessions using virtual, and at times offline mode in order to take stock of the problem and means to counter them.A dashboard should be created immediately, containing consolidated disease data from member countries. It would facilitate the exchange of disease information between member countries and help in planning effective control and preventive measures.Member nations should start a consultation/dialogue to achieve the immunological barrier so that the border areas can be immunised against TAD, which is crucial. SAARC and ILRI may act as a catalyst in bringing this to practice.SAARC in collaboration with international and regional partner agencies and centre of excellence on animal diseases and member states may devise a collaborative program on TAD surveillance, an early warning system, a tracking of animal movement, and vaccine synchronisation.Animal traders should be considered as a partner in TAD control through advocacy and confidence-building measures.

## 5. Way Forward

The TADs are posing a global and regional threat to livelihood, food and nutritional security, and trade due to globalisation of trade and informal movement of livestock including livestock products. The prevention and control mechanisms of TADs, at its source, are a global public good which requires coordinated efforts, solidarity, and the full political support from national and regional authorities. Ownership, leadership, and partnership are critical for regional collaboration, which demands mutual trust, confidence, and understanding among participating countries, particularly for information sharing.

Considering the evolving geopolitical situation in South Asia, it may not be possible to use the SAARC platform as desired. It will be strategic and realistic to initiate sub-regional cooperation among Bangladesh, Bhutan, India, and Nepal (BBIN), as they have extensive livestock movement and informal trade in cross-border areas and livestock, and poultry market value chain analysis were conducted in the past, which needs to be updated. Information sharing and cross-border surveillance of TAD should be prioritised, and it will take time to build mutual trust, confidence, and understanding to embark on such a mission.

The national veterinary services must have a strong and efficient technical team with a functional infrastructure, not only at the national level but also at sub-national levels [70]. SAARC countries should make best use of available operational tools, gap analysis, discussion-based and operations-based exercises such as the National IHR-PVS Bridging workshop, the operational tool on joint risk assessment, the multisectoral coordination mechanism, and surveillance and information sharing, as well as simulation exercises to test the functional status of preparedness and response, the contingency plan, interagency coordination, and the communication mechanism.

A network of laboratory diagnostic and referral services should be established and strengthened in South Asia. There are already regional reference laboratories for TADs identified under GF-TAD initiatives and OIE-Reference laboratories for avian influenza, and it is time to activate them by facilitating a material transport agreement (MTA) for smooth sample flow from neighbouring countries. They have to play a proactive role in providing external quality assurance and vaccine quality control as per OIE standards.

The quality and cost of vaccines are critical for disease control and subsequent eradication. It is desirable to have a regional vaccine bank for priority TADs, starting with PPR as a regional public good. There is a need for investment for innovation of new livestock and poultry vaccines and cost-effective diagnostic tools using nanotechnology through public–private partnership. India has been a global hub for innovation and technology transfer in the pharmaceutical and biotechnological industry sector, and it is time to motivate and incentivise them.

There should be a science-based government policy to import exotic breeds of livestock animals. Regulatory frameworks should be developed to enforce sanitary and phytosanitary (SPS) measures, including harmonisation of such measures among South Asian countries. There is complexity to perform risk analysis of potential introduction of TADs through import of genetic materials of exotic breeds such as semen, ova, and embryo due to the lack of data and information.

Cross-border collaboration for surveillance, early warning and animal movement control, and synchronisation of vaccination campaigns should be encouraged on a bilateral or multilateral basis for as many countries that share a porous border.

Involvement of national stakeholders and community engagement for surveillance, prevention, and control of regional priority TADs is critical, requiring rational utilisation of mass communication and social media networks. The lesson learnt from the infodemic during the COVID-19 pandemic should be taken into consideration in terms of the better understanding of public perception and action. Animal traders should be considered as partners in TAD control through advocacy and confidence-building measures.

## Figures and Tables

**Figure 1 vetsci-09-00586-f001:**
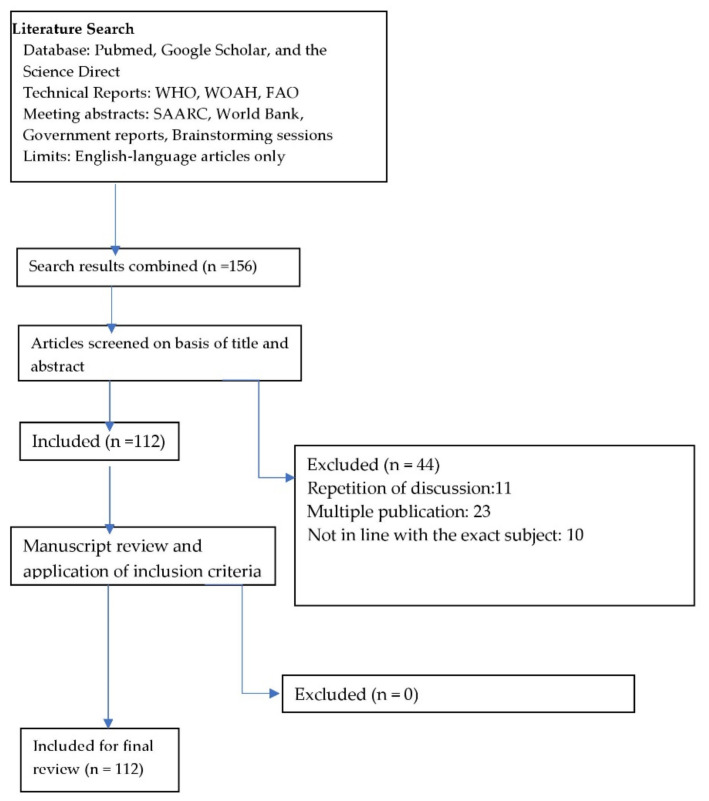
PRISMA statement flow diagram: summary of systematic search and review process.

**Figure 2 vetsci-09-00586-f002:**
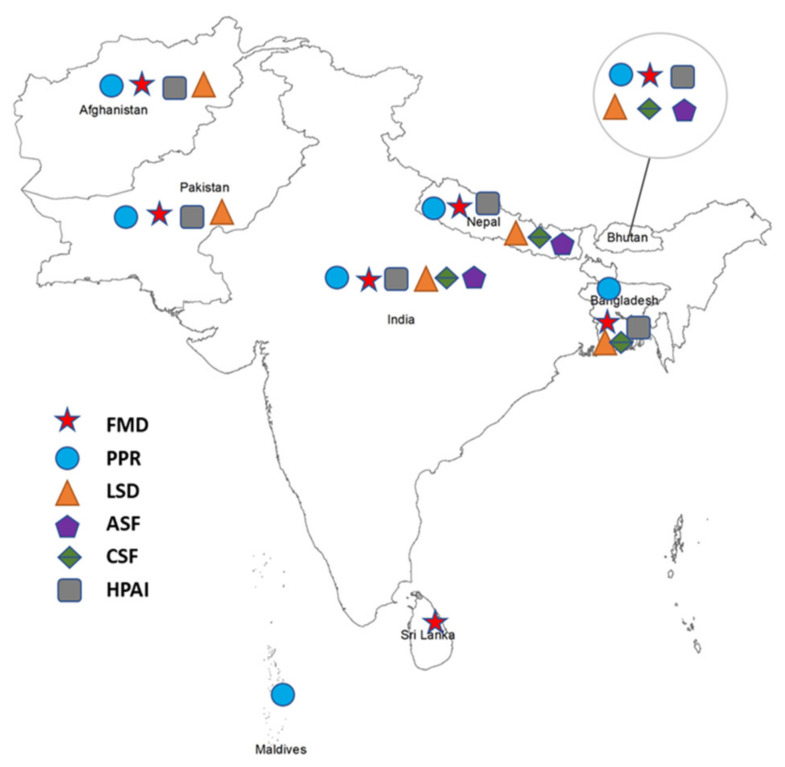
Status of six major TADs of viral origin in SAARC countries.

**Figure 3 vetsci-09-00586-f003:**
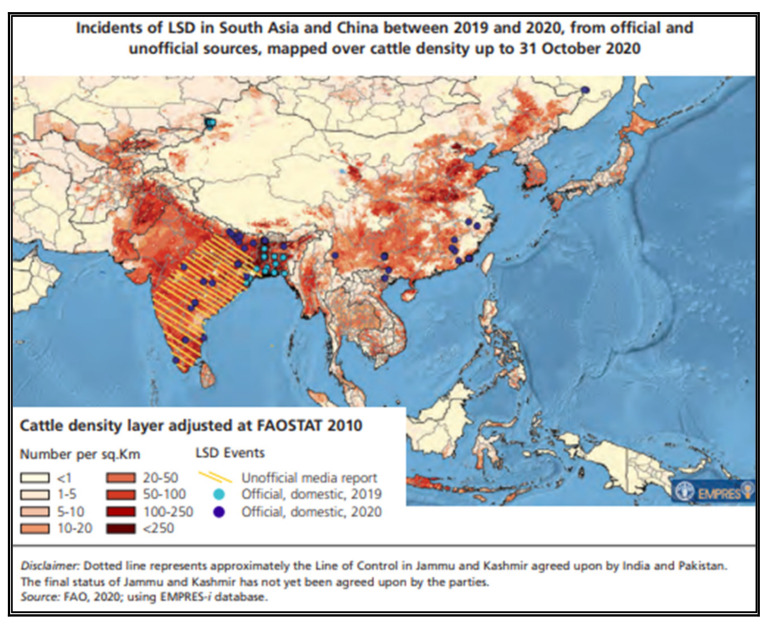
Qualitative risk assessment of LSD in South, East, and Southeast Asia.

**Figure 4 vetsci-09-00586-f004:**
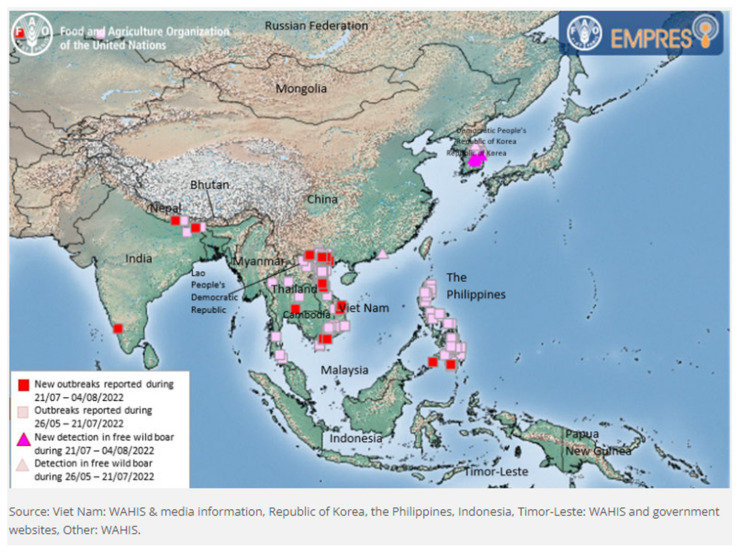
ASF situation in Asia (26 May–8 August 2022) [50].

**Figure 5 vetsci-09-00586-f005:**
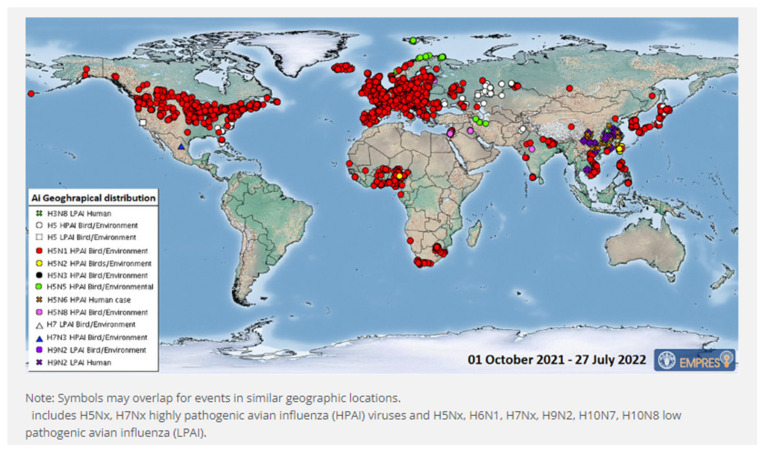
Global distribution of AIV with zoonotic potential observed since 1 October 2021 (i.e., current wave) [53].

**Figure 6 vetsci-09-00586-f006:**
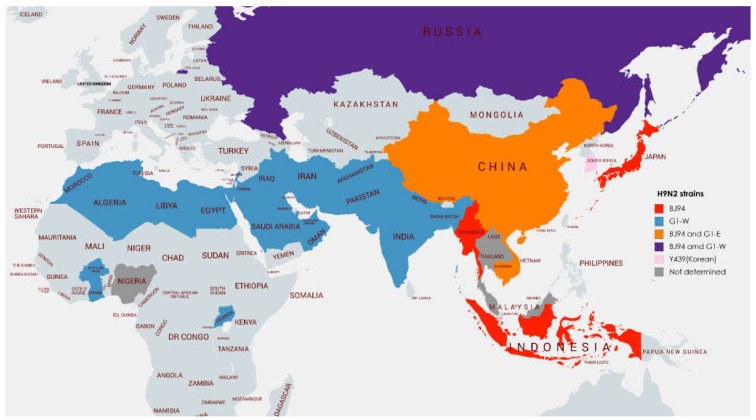
Phylogeographic range of poultry-adapted H9N2 lineages [55].

**Table 1 vetsci-09-00586-t001:** Updated PPR stage progression 2018–2030 based on country self-assessment [3].

Countries	2018	2019	2020	2021	2022	2023	2024	2025	2026	2027	2028	2029	2030
**Afghanistan ***	2	3						4		Free status			
**Bangladesh**	2			3			4		Free status				
**Bhutan**	2	3	4		Free status								
**India**	2					3		4		Free status			
**The Maldives**	2		3	4		Free status							
**Nepal**	1	2			3				4		Free status		
**Pakistan**	2			3				4		Free status			
**Sri Lanka**	1	4		Free status									

* Based on 2016 situation.

**Table 2 vetsci-09-00586-t002:** Cumulative number of confirmed human cases for AI A(H5N1) reported to the WHO, 2003–2021 [54].

Countries	2003–2009		2010–2014		2015–2019		2020		2021		Total	
	Cases	Deaths	Cases	Deaths	Cases	Deaths	Cases	Deaths	Cases	Deaths	Cases	Deaths
Bangladesh	1	0	6	1	1	0	0	0	0	0	8	1
India	0	0	0	0	0	0	0	0	1	1	1	1
Nepal	0	0	0	0	1	1	0	0	0	0	1	1
Pakistan	3	1	0	0	0	0	0	0	0	0	3	1
Others	464	281	227	124	158	47	1	0	0	0	850	452
Total	468	282	233	125	160	48	1	0	1	1	863	456

**Table 3 vetsci-09-00586-t003:** Structure and functional system under regional GF-TADs, South Asia.

Sub-Region	South Asia		
RSO	SAARC		
RSU	Kathmandu		
Priority diseases	HPAI	FMD	PPR
Leading laboratory	Pakistan (National Reference Laboratory for Poultry Disease, Islamabad)	India (Directorate on FMD, Mukteshwar)	Bangladesh (Bangladesh Livestock Research Institute, Dhaka)
Roadmap (strategy)	FAO regional HPAI strategy	FMD_PCP roadmap for SAARC	-
Epi network (details)	Established (within SAARC RSU, FAO, Kathmandu)		
Lab network (details)	Established (within SAARC RSU, FAO, Kathmandu)		

**Table 4 vetsci-09-00586-t004:** Status of PVS evaluation and PVS gap analysis, South Asia, 2008–2019.

SAARC Countries	PVS Evaluation	PVS Gap Analysis	PVS Follow-Up Mission	Remarks
Afghanistan	2010		2017	Vet. Legislation (2010)
Bangladesh	2011	2015		
Bhutan	2008	2009/2015	2015	Vet. Legislation (2012)/Laboratory (2016)
India	2018			
The Maldives	2011			
Nepal	2008	2011		
Pakistan	2014	2019		Vet. Legislation
Sri Lanka	2008	2011		

Source: the World Organisation for Animal Health.

## Data Availability

Not applicable.

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
