# Peer review of "An Overview of Transboundary Animal Diseases of Viral Origin in South Asia: What Needs to Be Done?"

_vetsci, 2022, doi:10.3390/vetsci9110586_

Round 1

Reviewer 1 Report

Introduction is confused and poor of relevant references, it is very difficult to understand the connection with the main aim of this work. Furthermore, which is the aim? This must be define at the end of the introduction in order to explain what the authors would like to demonstrate or describe. I strongly suggest to rewrite avoiding less important information or sentences (i.e., It will be unfair if we do not mention Rinderpest which was behind the introduction of modern veterinary practice in south Asia and Rinderpest eradication is a proof of concept that TADs can be eradicated from this region through coordinated actions and international partnership) and focus on essential information. Include the aim of the work is mandatory. Please, check that each acronimous has been defined first time you use it (i.e., FMD, PPR, GDP), and avoid to define them more than one time (i.e., SAARC). Avoid to speak as first person (line 45: we have included eight SAARC countries as part of South Asia). Move the figure 1 into methods or results.

M&M: if the aim of the work is to perform a review, first the Prisma must be compiled. Furthermore, all criteria applied to search and select the paper must be elicited. Absolutely avoid sentences as: "The authors have worked on TADs surveillance, prevention and control during their government service in respective countries and participated in brainstorming discussion on regional and national policy and strategy for control and eradication of priority TADs from time to time. Their professional thought, judgement and expertise have been reflected in writing the manuscript". This is not a curriculum vitae.

The paper must be completely reformulated following the scientific aim of a review. results and discussion cannot be evaluated if material and methods are reformulated previously.

Reviewer 2 Report

Dear Authors: You have done good job in compiling several existing resources with the focus on the current TADs in the region.  The report, however is so detailed and it can be reduced significantly by avoiding what was addressed by the various previous reports and publications.  As per your title  what is needed is the unique contribution of this submission.   However, you did not specify the plan of action or a set of recommendations after you listed in details the determinants of the problem.   I believe such recommendations will be the most useful contribution to the scientific community and animal health authority in general. 

My suggestion, therefore, is to reduce the length of the background and elaborations on the history and current status of these diseases and to add few sentences (2-3 paragraphs) of your recommendations for moving forward that are supported with evidence.  

I wish you the best if you decide to revise the submission. 

Sincerely

Author Response

Dear Authors: You have done good job in compiling several existing resources with the focus on the current TADs in the region.  The report, however is so detailed and it can be reduced significantly by avoiding what was addressed by the various previous reports and publications. 

Needful done. Repetition of the results from previous reports and publications is removed.

As per your title  what is needed is the unique contribution of this submission.   However, you did not specify the plan of action or a set of recommendations after you listed in details the determinants of the problem.   I believe such recommendations will be the most useful contribution to the scientific community and animal health authority in general.

The main aim of the paper is mentioned at the end of the introduction. The plan of action/set of recommendations are listed in detail.

My suggestion, therefore, is to reduce the length of the background and elaborations on the history and current status of these diseases and to add few sentences (2-3 paragraphs) of your recommendations for moving forward that are supported with evidence. 

Needful done. The length of the background section is reduced; The recommendations for moving forward are listed in detail.

Round 2

Reviewer 1 Report

The authors did not thake into account all my suggestions. I repeat: check the acrinomous and do not repert them in extension format in alla chapters. Furthermore, is not cleare the origin of the paper selected. Plese, use the PRISMA format, as need for each review.

Furthermore, the paper is very too long and boring. Drastically reduce each session.

Author Response

Point 1: Check the acronymous and do not repeat them in extension format in all chapters.

Response 1: Acronymous are reviewed properly and it is not repeated again in extension format in all chapters. 

Point 2: Furthermore, is not clear the origin of the paper selected. Please, use the PRISMA format, as needed for each review.

Response 2: PRISMA format is used and mentioned under materials and methods.

Point 3: Furthermore, the paper is very too long and boring. Drastically reduce each session.

Response 3: Each session has been shortened, and only the findings of priority have been kept.

Point 4: English language, the following link can be used as a reference

Response 4: The paper had been proofread and the english language is improved.

Reviewer 2 Report

Thanks for considering the suggestions 

Author Response

No comment received. 

Round 3

Reviewer 1 Report

Please, check the acronymous. I'me very tired to repeat...

TDS is define in the abstract, not in the main text, as well as FMD. FAO is not specified.